# Risk of Burning Mouth Syndrome in Patients with Migraine: A Nationwide Cohort Study

**DOI:** 10.3390/jpm12040620

**Published:** 2022-04-11

**Authors:** Dong-Kyu Kim, Hyun-Joo Lee, Il Hwan Lee, Jae-Jun Lee

**Affiliations:** 1Institute of New Frontier Research, Division of Big Data and Artificial Intelligence, Chuncheon Sacred Heart Hospital, Hallym University College of Medicine, Chuncheon 24253, Korea; leekul79@gmail.com; 2Department of Otorhinolaryngology-Head and Neck Surgery, Chuncheon Sacred Heart Hospital, Hallym University College of Medicine, Chuncheon 24253, Korea; entkim@hallym.or.kr; 3Department of Anesthesiology and Pain Medicine, College of Medicine, Chuncheon Sacred Heart Hospital, Hallym University College of Medicine, Chuncheon 24253, Korea

**Keywords:** burning mouth syndrome, migraine, cohort study, oral cavity

## Abstract

Migraine is a common neurological disease that causes a variety of symptoms, most notably throbbing, which is described as a pulsing headache on one side of the head. Burning mouth syndrome (BMS) is defined as an intra-oral burning sensation. Currently, no medical or dental cause has been identified for BMS. Interestingly, neuropathic pain is a characteristic feature of BMS; however, it remains unclear whether migraine can cause BMS. We aimed to identify the association of migraine with the risk of developing BMS. We used a representative nationwide cohort sample of approximately 1 million patients from 2002 to 2013 to investigate the prospective association between migraine and BMS. A total of 4157 migraine patients (migraine group) and 16,628 patients without migraine (comparison group) were enrolled after 1:4 propensity score matching. The overall incidence of BMS was significantly higher in the migraine group (0.15 per 1000 person-years) than in the comparison group (0.05 per 1000 person-years). The adjusted HR for patients with migraine who reported BMS events during the 10-year follow-up period was 2.96 (95% confidence interval, 1.02–8.56), after adjusting for other covariates. However, in the subgroup analysis, the adjusted HR for BMS events did not show a significant difference between the migraine and comparison group according to sex, age, and comorbidities. This study suggests that migraine is associated with an increased incidence of BMS. Therefore, clinicians should be attentive to detect BMS at an early stage when treating patients with migraine.

## 1. Introduction

Migraine is one of the most common neurological disorders, characterized by episodes of severe, and often unilateral, throbbing or pulsating headaches associated with multiple symptoms such as nausea, photophobia, and phonophobia; approximately 15% of the general population has experienced a migraine [1]. It is the most debilitating form of headache and causes substantial levels of disability [1]. Additionally, common comorbidities of migraine include neck pain and psychological illnesses, such as depression and anxiety, all of which are leading causes of disability worldwide [2,3]. The pathophysiology of migraine is not completely understood; however, recent studies have proposed that some neuropeptides, such as calcitonin gene-related peptide and substance P, act as mediators [4,5,6]. Thus, neuropathy may be involved in patients with migraine.

Burning mouth syndrome (BMS) is an idiopathic chronic pain disorder characterized by a persistent burning sensation in the oral cavity, for which no medical or dental cause is yet known [7,8]. The overall prevalence of BMS is approximately 4%, it is mainly observed in middle-aged or older adult women, and usually presents as an intense burning sensation of the tongue or oral mucosa [9]. Primary BMS is caused by damage to the nerves that control pain and taste, whereas secondary BMS is usually caused by an underlying medical condition. Although the primary cause of BMS remains unclear, several etiological factors, such as local disorders (orodental diseases, bacterial and fungal infections, and salivary gland disorders), systemic disorders (folate deficiency, diabetes, thyroid disease, and Sjögren’s syndrome), and psychiatric disorders (depression, anxiety, and obsessive compulsive disorder), have been proposed to contribute to its pathogenesis [9,10]. Moreover, accumulating evidence shows that psychological factors and neuropathy play an important role in the development of BMS [11,12,13,14]. Although psychological factors and neuropathy are often combined with migraine, to date, no study has investigated the relationship between migraine and BMS. Additionally, epidemiological studies on chronic pain have revealed that pain sites are often multiple and coexistent in the same patient [15,16]. Generally, chronic overlapping pain conditions represent a co-aggregation of widespread pain disorders, consisting of not only temporomandibular disorders, fibromyalgia, irritable bowel syndrome, chronic tension-type headache, chronic lower back pain, chronic pelvic pain, but also migraine and BMS.

Given these observations, we conducted a retrospective cohort study using the insurance data of a nationwide population, and evaluated the risk of BMS in patients with migraine. This dataset from a representative nationwide cohort allowed us to comprehensively trace the history of medical services availed by the patients. Further, it provided a unique opportunity to examine the association between migraine and risk of developing BMS, while adjusting for clinical and demographic factors.

## 2. Materials and Methods

This retrospective nationwide cohort study with propensity score matching used data from the National Health Claims Database collected by the South Korea National Health Insurance Service (KNHIS). The study protocol was approved by the Institutional Review Board of Hallym Medical University, Chuncheon Sacred Hospital (IRB No. 2021-08-006), and the study adhered to the tenets of the Declaration of Helsinki. The requirement for written informed consent was waived by the Institutional Review Board of Hallym Medical University, Chuncheon Sacred Hospital because the KNHIS–National Sample Cohort (KNHIS–NSC) database used in this study comprised de-identified secondary data.

### 2.1. Study Population

We retrospectively chose a representative sample of 1,025,340 patients from the KNHIS–NSC database, who were registered between 2002 and 2013, and we identified patients older than 20 years with migraine using the diagnostic code G43 (International Classification of Diseases-10, ICD-10). In the KNHIS, all disease diagnostic codes were identified using the Korean Classification of Disease, fifth edition (KCD-5), a modification of the International Classification of Disease and Related Health Problems, 10th revision (ICD-10). In this study, we conducted a washout period of one year (2002) to remove the possibility of BMS diagnosis prior to migraine diagnosis. Additionally, to enhance the accuracy of diagnosis, we selected patients who were diagnosed with migraine (G43) more than twice between 2003 and 2005. Moreover, we excluded patients who (1) were diagnosed with BMS before the first diagnosis of migraine, (2) died due to any cause between 2002 and 2005, and (3) died after 2005 as a result of any accident. For the comparison group, consisting of individuals without migraine, we randomly selected propensity score-matched participants from the remaining cohort registered in the database, i.e., four participants without migraines for each patient with migraine. To enhance the power of the comparison group, we excluded patients who had been diagnosed with any headache-related diseases between 2002 and 2013. Each patient was followed up until 2013 or to the time of diagnosis for BMS (K14.6). Finally, a total of 4157 eligible patients were enrolled in the migraine group, and 16,628 patients in the comparison (non-migraine) group, for this study (Figure 1).

### 2.2. Outcome Variables

The covariates in this study included sex, age, residence, income level, and comorbidities. The study population was divided into three categories by age (<45, 45–64, and >64 years), residential area (Seoul, the largest metropolitan region in South Korea; other metropolitan cities in South Korea; and small cities and rural areas in South Korea), and income level (≤30%, around 30.1–69.9%, and above ≥70% of the median income of the study group). Specifically, we analyzed comorbidities using the Charlson comorbidity index (CCI). Death and incidence of BMS were used as the primary study endpoints. Patients who did not experience BMS or those who were alive until 31 December 2013 were excluded after this time point (Appendix A Table A1).

### 2.3. Statistical Analyses

We measured the incidence using a person-years method which calculated the duration for each patient between enrollment date and primary endpoint. The incidence rate was also expressed as per 1000 person-years. To identify whether migraine increased the risk of occurrence of specific diseases, we used Cox proportional hazard regression analysis to calculate the hazard ratio (HR) and 95% confidence interval (CI). All statistical analyses were performed using R (version 3.4.3; R Foundation for Statistical Computing, Vienna, Austria), with a significance level of *p* = 0.05.

## 3. Results

The characteristics of each cohort dataset are listed in Table 1. We observed a similar distribution of each variable between the two groups (Figure 2); that is, the group matching based on sex, age, residence, income level, and comorbidities was appropriate. A total of 39,371.21 person-years in the migraine group, and 162,034.51 person-years in the comparison group, were calculated for BMS. The overall incidence of BMS was significantly higher in the migraine group (0.15 per 1000 personyears) than in the comparison (non-migraine) group (0.05 per 1000 person-years). 

We also analyzed the HR for development of BMS during the 10-year follow-up period using univariate and multivariate Cox regression models (Table 2). After adjusting for sex, age, residence, income level, and comorbidities, we found that migraine was significantly associated with the development of BMS (adjusted HR = 2.96, 95% CI: 1.02–8.56). However, no significant association was identified between any of the other factors analyzed (sex, age, residence, income level, comorbidities) and the development of BMS events. 

Kaplan–Meier survival curves with the log-rank test results for the 10-year follow-up period are presented in Figure 3, which depicts the cumulative incidence of BMS in the migraine and comparison groups. The risk of BMS was significantly higher in the migraine group than in the comparison group (log-rank *p* < 0.001). The results of the log-rank tests indicated that subjects with BMS were more frequent in the migraine group than in the comparison (non-migraine) group.

Next, we analyzed the HR for BMS events in the migraine and non-migraine groups according to sex, age, and comorbidities (Appendix A Table A2, Table A3 and Table A4). However, we did not detect any significant difference in the adjusted HR for BMS development between the migraine and comparison group according to sex, age, and comorbidities.

## 4. Discussion

In this study, we investigated whether migraines could increase the risk of BMS events. For this analysis, we selected participants who were matched for sociodemographic factors from a nationwide 10-year longitudinal cohort database of 1,025,340 South Korean patients. To the best of our knowledge, this study is the first to analyze the risk of developing BMS events in patients with migraine. Interestingly, we found a significant difference between the migraine and comparison groups in the number of patients who developed BMS. After adjusting for sociodemographic characteristics and comorbidities, we found that the patients with migraine had a 2.96 times higher risk of developing BMS than those without migraines. Moreover, the log-rank test results indicated that patients with migraine developed BMS more frequently than patients without migraine during the 10-year follow-up period.

In the present study, our hypothesis was that psychological factors and neuropathy may be the possible mechanisms for the association of migraine with subsequent BMS development. Migraine is a multifactorial disorder with genetic, hormonal, environmental, and psychological aspects. It is usually associated with a wide range of psychiatric comorbidities. Generally, psychiatric comorbidities can increase susceptibility to pain by increasing transmission to peripheral nerve receptors [17,18,19]. Among these psychiatric comorbidities, anxiety and depression are strongly associated with migraines [20,21,22]. Additionally, if these comorbid psychiatric conditions are left untreated, the risk of migraine becoming a chronic disorder increases [22]. Further, high levels of anxiety and depression are also prevalent in BMS patients. Several studies have described that BMS patients demonstrate more anxious and depressed mood states compared to control groups [11,23,24,25]. Thus, BMS is similar to other chronic pain conditions, where high rates of sub-syndromal psychological disorders have been observed. Given these findings, we thought that psychological components may be one of the links between migraine and BMS.

The exact mechanism underlying migraine pathogenesis is poorly understood. Currently, the neurovascular hypothesis is widely accepted as the mechanism underlying headaches in migraine [26,27]. This hypothesis states that the pain in migraine originates from the trigeminovascular system—activation of the trigeminal sensory nerves releases several vasoactive neuropeptides, including calcitonin gene-related peptide (CGRP), neurokinin A, and substance P. Release of these vasoactive neuropeptides triggers cerebral vasodilation and dural plasma extravasation, leading to neurogenic inflammation. Hence, CGRP is being used as a new therapeutic target in migraine patients [6]. The etiology of BMS remains poorly understood, which makes its treatment quite challenging. However, there are several clues indicating that neuropathy could contribute to the development of BMS [9,28,29,30,31]. These studies showed that damage to peripheral small nerve fibers can result in clinical presentations often described as burning, tingling, and numbness, and that, compared with healthy controls, BMS patients tend to have a decreased tolerance to a painful heat stimulus at the tip of the tongue. In addition, small nerve fiber neuropathy along with a significant reduction in small fibers’ density in the pain areas of BMS patients has been observed. Moreover, some studies have shown that approximately 30–60% of BMS patients experience neuropathic pain [32,33]. Other studies have also reported that systemic inflammation affects the upregulation of local inflammation [34]. Thus, in addition to a psychological component, we suggest that the increased neurogenic inflammation could contribute to neuropathic injury, which may induce the development of BMS.

Our study had some unique strengths. First, our dataset was based on a nationwide population and has been recently used in multiple published studies [35,36,37,38,39]. Second, to enhance the accuracy of operation definition for BMS and migraine, we selected BMS patients diagnosed by otorhinolaryngologists, and migraine patients diagnosed by neurologists. Third, our study had a relatively long follow-up period, and was adjusted for most major comorbidities using CCI, which categorizes the comorbidities of the patients based on the ICD diagnosis codes present in the KNHIS–NSC database. Finally, previous studies have confirmed the reliability of the prevalence of 20 major diseases in the KNHIS–NSC database [40,41].

This study also had some limitations. First, we could not access any specific personal health data, such as smoking history and alcohol consumption. Therefore, we could not adjust for these potential confounding factors. Second, instead of using a diagnosis based on medical history, physical examination results, and laboratory findings, we solely conducted the diagnosis of migraine and BMS based on the ICD-10. Thus, it may be less accurate and could have created potential bias in classification while assigning the participants to the groups in our study. For example, we could not differentiate between primary and secondary BMS. To overcome this problem, we included only those patients with migraine or BMS who were diagnosed by neurologists or otorhinolaryngologists, respectively. Third, there is a wide spectrum of migraines, depending on the phenotype of the disease, which could not be distinguished in this study based on the simplistic diagnostic code system used. Fourth, we could not determine whether the age at migraine onset corresponded with age at the first hospital visit for the migraine. In addition, our registry does not provide information regarding the frequency and severity of the migraine; therefore, we could not investigate whether these factors may have had a differential effect on the risk of BMS. Fifth, we could not consider the effect of anti-migraine medication because we did not have the data on medication compliance for each patient. To overcome these limitations, we matched the migraine and non-migraine groups using propensity scores. Finally, this study did not investigate the pathophysiologic mechanisms underlying the relationship between migraine and BMS. Thus, future clinical studies are necessary to provide stronger evidence for the link between migraine and BMS.

In conclusion, this study examined the association between migraine and the risk of BMS events, while adjusting for clinical and demographic factors. We found an increased risk of BMS events in patients with migraine than in those without migraines. Therefore, clinicians should be aware of the potential development of BMS in patients with migraine and provide these patients with additional therapies to reduce the risk of developing BMS.

## Figures and Tables

**Figure 1 jpm-12-00620-f001:**
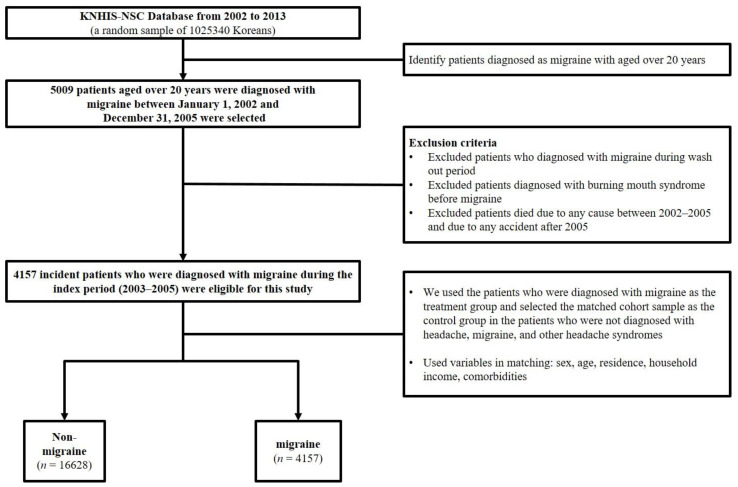
Schematic representation of the study design.

**Figure 2 jpm-12-00620-f002:**
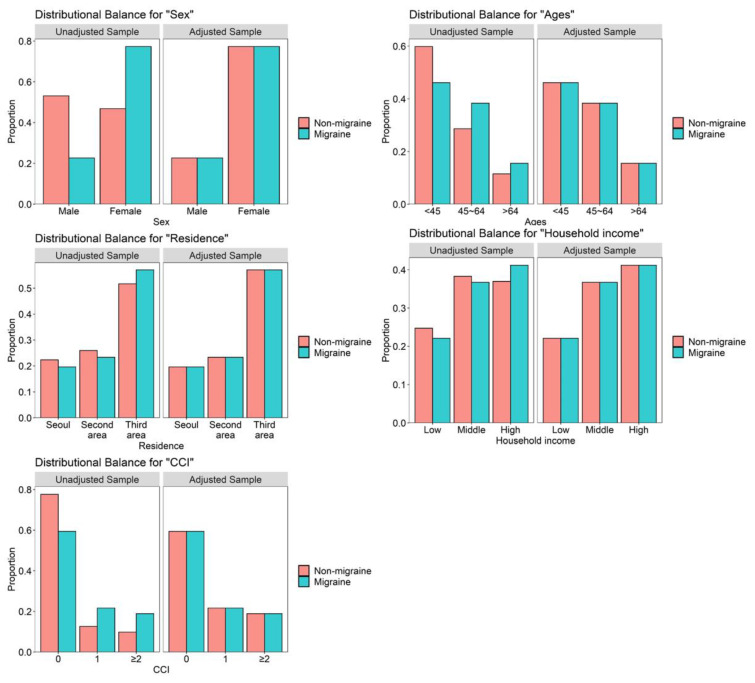
Balance plots for five variables before and after propensity score matching.

**Figure 3 jpm-12-00620-f003:**
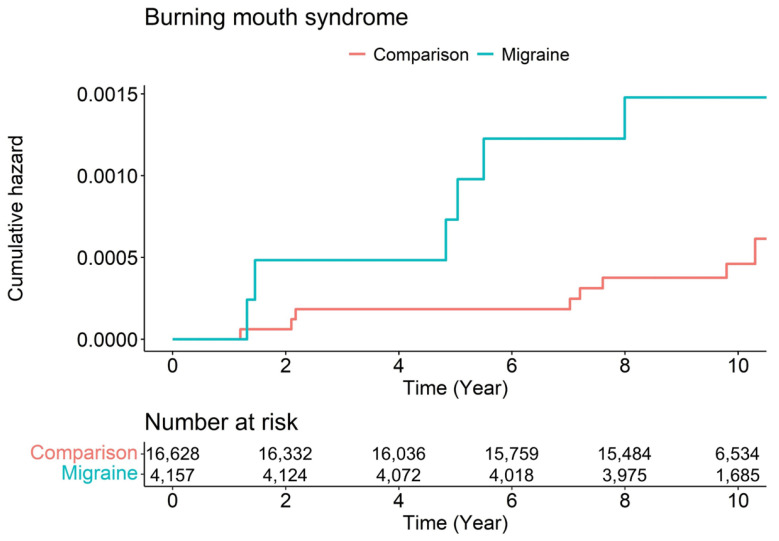
Kaplan–Meier survival curves and log-rank tests for the pathogenesis of burning mouth syndrome between the comparison and migraine groups.

**Table 1 jpm-12-00620-t001:** Characteristics of the cohort.

Variables	Comparison (*n* = 16,628)	Migraine (*n* = 4157)	*p* Value
**Sex**			1000
Men	3772 (22.7%)	943 (22.7%)	
Women	12,856 (77.3%)	3214 (77.3%)	
**Ages (years)**			1000
<45	7672 (46.1%)	1918 (46.1%)	
45–64	6376 (38.3%)	1594 (38.3%)	
>64	2580 (15.5%)	645 (15.5%)	
**Residence**			1000
Seoul	3264 (19.6%)	816 (19.6%)	
Second area	3880 (23.3%)	970 (23.3%)	
Third area	9484 (57.0%)	2371 (57.0%)	
**Household income**			1000
Low (0–30%)	3676 (22.1%)	919 (22.1%)	
Middle (30–70%)	6104 (36.7%)	1526 (36.7%)	
High (70–100%)	6848 (41.2%)	1712 (41.2%)	
**CCI**			1000
0	9880 (59.4%)	2470 (59.4%)	
1	3604 (21.7%)	901 (21.7%)	
≥2	3144 (18.9%)	786 (18.9%)	

Comparison, subjects without migraines; Seoul, the largest metropolitan area; second area, other metropolitan cities; third area, other areas; CCI, Charlson comorbidity index.

**Table 2 jpm-12-00620-t002:** The overall incidence per 1000 person-years and HR (95% CI) of burning mouth syndrome.

Variables	N	Case	Person-Year	Incidence	Unadjusted HR (95% CI)	Adjusted HR (95% CI)	*p* Value
**Group**							
Comparison	16,628	8	162,034.51	0.05	1.00 (ref)	1.00 (ref)	
Migraine	4157	6	39,371.21	0.15	3.10 (1.08–8.97)	2.96 (1.02–8.56)	0.045
**Sex**							
Men	4715	2	44,313.74	0.05	1.00 (ref)	1.00 (ref)	
Women	16,070	12	157,091.98	0.08	1.68 (0.38–7.52)	1.65 (0.36–7.44)	0.516
**Age (years)**							
<45	9590	1	96,319.54	0.01	1.00 (ref)	1.00 (ref)	
45–64	7970	7	78,831.25	0.09	8.55 (1.05–69.46)	7.39 (0.90–60.59)	0.062
>64	3225	6	26,254.93	0.23	22.47 (2.70–186.89)	15.04 (1.75–128.92)	0.013
**Residence**							
Seoul	4080	3	40,086.03	0.07	1.00 (ref)	1.00 (ref)	
Second area	4850	1	47,235.26	0.02	0.28 (0.03–2.73)	0.28 (0.03–2.68)	0.268
Third area	11,855	10	114,084.43	0.09	1.17 (0.32–4.26)	0.94 (0.26–3.44)	0.928
**Household income**							
Low	4595	6	43,909.28	0.14	1.00 (ref)	1.00 (ref)	
Middle	7630	4	74,159.59	0.05	0.39 (0.11–1.40)	0.50 (0.14–1.79)	0.288
High	8560	4	83,336.85	0.05	0.35 (0.10–1.24)	0.39 (0.11–1.38)	0.143
**CCI**							
0	12,350	4	121,791.34	0.03	1.00 (ref)	1.00 (ref)	
1	4505	5	43,961.34	0.11	3.46 (0.93–12.90)	2.44 (0.64–9.23)	0.190
≥2	3930	5	35,653.04	0.14	4.28 (1.15–15.92)	2.98 (0.79–11.27)	0.108

Seoul, the largest metropolitan area; second area, other metropolitan cities; third area, other areas; CCI, Charlson comorbidity index; HR, hazard ratio; CI, confidence interval.

## Data Availability

The authors confirm that the data supporting the findings of this study are available within the article.

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
