# Peer review of "Risk of Burning Mouth Syndrome in Patients with Migraine: A Nationwide Cohort Study"

_jpm, 2022, doi:10.3390/jpm12040620_

Round 1
Reviewer 1 Report
The manuscript is of interest and as it investigates novel clinical facets of oral chronic pain of potential interest for further investigations.
Below, the comments for authors.
- The article is neatly written.
- I request to add a paragraph that describes the association between BMS and others chronic painful disorders (e.g. fibromyalgia, urologic chronic pelvic pain syndrome): what the epidemiologic evidence of such an association is. I may suggest the authors can look up and cite this recent article on this topic:
Crocetto F, Coppola N, Barone B, Leuci S, Imbimbo C, Mignogna MD. The association between burning mouth syndrome and urologic chronic pelvic pain syndrome: A case-control study. J Oral Pathol Med. 2020 Sep;49(8):829-834. doi: 10.1111/jop.13097. Epub 2020 Aug 30. PMID: 32797728.
- A few grammar mistakes require the manuscript to be revised linguistically.
- Discussion section: Since this study is the first time to propose the association between BMS and migraine on a large sample size, I hope that the author can put forward some reasonable hypotheses to explain the specific reasons and mechanism of this association (see line 160-161)
Author Response
The manuscript is of interest and as it investigates novel clinical facets of oral chronic pain of potential interest for further investigations.
Below, the comments for authors.
The article is neatly written.
I request to add a paragraph that describes the association between BMS and others chronic painful disorders (e.g. fibromyalgia, urologic chronic pelvic pain syndrome): what the epidemiologic evidence of such an association is. I may suggest the authors can look up and cite this recent article on this topic: Crocetto F, Coppola N, Barone B, Leuci S, Imbimbo C, Mignogna MD. The association between burning mouth syndrome and urologic chronic pelvic pain syndrome: A case-control study. J Oral Pathol Med. 2020 Sep;49(8):829-834. doi: 10.1111/jop.13097. Epub 2020 Aug 30. PMID: 32797728.
Answer: Thank you for your kind comment. As you recommened, we added the paragraph and citation as follows: “Additionally, epidemiological studies for chronic pain revealed that pain sites are often multiple and coexistent in the same patient [15–16]. Generally, chronic overlapping pain conditions that represent a co-aggregation of widespread pain disorders consist of not only temporomandibular disorders, fibromyalgia, irritable bowel syndrome, chronic tension-type headache, chronic lower back pain, chronic pelvic pain, but also migraine and BMS.”
A few grammar mistakes require the manuscript to be revised linguistically.
Answer: We thoroughly checked for grammatical issues, again.
Discussion section: Since this study is the first time to propose the association between BMS and migraine on a large sample size, I hope that the author can put forward some reasonable hypotheses to explain the specific reasons and mechanism of this association (see line 160-161)
Answer: We agreed with your suggestion. Accordingly, we added more descriptions regarding the possible mechanism of this association.
Reviewer 2 Report
Dear Editor,
This study investigates the association between migraine and the risk of burning mouth syndrome (BMS) events while adjusting for clinical and demographic factors. It suggests that there is an increased risk of BMS events in patients with migraine than in those without migraines.
Please find below my concerns.
- First of all, there is no satisfying discussion about the pathophysiology of BMS and migraine. Why BMS is more common in migraine patients?
- Another thing is the time lag between migraine attacks onset and BMS onset. If there is a significant time lag between them then there should be sensitization of nociceptors inducing BMS. Please give the time lag, and discuss this data regarding the pathophysiology of BMS seeing in migraine patients.
Best regards,
Author Response
Dear Editor,
This study investigates the association between migraine and the risk of burning mouth syndrome (BMS) events while adjusting for clinical and demographic factors. It suggests that there is an increased risk of BMS events in patients with migraine than in those without migraines.
Please find below my concerns.
First of all, there is no satisfying discussion about the pathophysiology of BMS and migraine. Why BMS is more common in migraine patients? Another thing is the time lag between migraine attacks onset and BMS onset. If there is a significant time lag between them then there should be sensitization of nociceptors inducing BMS. Please give the time lag, and discuss this data regarding the pathophysiology of BMS seeing in migraine patients.
Answer: Firstly, thank you for your kind review. We agreed with your observations and accordingly, we added more descriptions regarding the possible mechanism of this association. In this study, we also detected the change in risk ratio for BMS events according to the time lag and have presented this in Figure 3 as the cumulative hazard ratio based on year. As Figure 3 shows, our findings indicated the increased subsequent development of BMS in a time-dependent manner.
Reviewer 3 Report
The authors published a retrospective analysis of patient-data from the KNHIS database (South Korea). They analyze the risk of burning moth syndrome (BMS) in association to migraine. The overall incidence of BMS was higher in the migraine group (0.15 vs. 0.05 per 1000 person-years). An adjusted HR during a 10year follow-up period for patients with migraine who reported BMS event was 2.96. The subgroup analysis show no significant difference. The authors suggested that migraine was associated with an increase incidence of BMS.
In Short: This retrospective analysis from a Korean National Health Service database reveals a possible association between BMS and migraine. It is a possible statistical association, although it remains unclear whether there is a direct pathophysiological relationship between the two conditions. In addition, the association between migraine and other pain disorders is well known. What is striking here is that in other epidemiological work on burning mouth syndrome, a significantly increased prevalence has been reported, particularly in older patients and here especially in women older than 60 years and in menopause. In a review of all age groups, the prevalence was approximately 0.01%, whereas in the present work it was only 0.07% (Jaääskeläinen SK et al. 2017). Overall, the epidemiological data on burning mouth syndrome and migraine are not presented at all in the paper; this would be necessary to better assess the results. In addition, no work is presented that provides good evidence that migraine alone is associated with a higher risk of anxiety and depression, back pain, and somatoform disorder (Gerstl L et al. 2021). Since some of these factors are also considered risk factors for burning mouth syndrome, it must at least be discussed here how the authors evaluate their data in relation to this. Furthermore, it is unclear why 852 patients with migraine were excluded from the study (Figure 1), which may significantly influence the results. In addition, the Charlson comorbidity index was used to assess possible comorbidities. However, this index does not capture most of the risk factors for secondary burning mouth syndrome. Furthermore, it is unclear which diagnostic criteria were used to diagnose migraine or polymorphic syndrome. Furthermore, no distinction was made between primary and secondary burning mouth syndrome.
I conclude that the manuscript in this version with major revisions adaptable for publication.
Comments to the authors:
Abstract
The first sentence is misleading because the relationship between neuropathic pain and migraine is unclear. The whole abstract should not discuss a direct relationship between Burning mouth syndrome and migraine but rather a possible association between the two conditions.
Introduction
Epidemiological data on burning mouth syndrome and migraine are lacking here. In addition, data on the association between psychiatric disorders and burning mouth syndrome but also migraine should be presented. This is particularly important as it is a relevant factor in both. Furthermore, it should be made clear where the difference between primary and secondary burning mouth syndrome lies. In conclusion, again only a possible association between both disorders should be mentioned.
Material and methods
Did the classification of BMS and migraine follow ICHD criteria? Please indicate which valid classification was used. Furthermore, there is no description of how the diagnoses were collected from the data and who finally made the diagnosis. There is a lack of differentiation between primary and secondary BMS. Why were 852 patients with migraine excluded, this is only shown in Figure 1, but not justified in the text. In addition, it should be explained why the CCI was used as a score for comorbidities, since it does not include most comorbid diseases that are relevant for burning mouth syndrome and migraine (e.g. rheumatic diseases).
Results
It would also be useful to present the prevalence as well as incidence of burning mouth syndrome in the entire cohort. It would be useful to present the significant results in particular.
Discussion
There need not be a direct association between burning mouth syndrome and migraine; it is also possible that psychiatric disorders (depression and anxiety), which are very common and increase with age, are responsible for the suspected association.
The part on the association between migraine and burning mouth syndrome regarding systemic inflammation is not understandable
On the strengths of the study, here it is just unclear how the diagnosis of burning mouth syndrome of migraine was made in each case. Who made the diagnosis should be presented in the methods. Also here it is not clear why the CCI was selected and which connection should
Author Response
Comments to the authors:
Abstract
The first sentence is misleading because the relationship between neuropathic pain and migraine is unclear. The whole abstract should not discuss a direct relationship between Burning mouth syndrome and migraine but rather a possible association between the two conditions.
Answer: We agree with your opinion. Thus, we modified this sentence as follows: “Migraine is a common neurological disease that causes a variety of symptoms, most notably throbbing, which is described as a pulsing headache on one side of the head. Burning mouth syndrome (BMS) is defined as an intra-oral burning sensation. Currently, no medical or dental cause has been identified for BMS. Interestingly, neuropathic pain is a characteristic feature of BMS; however, it remains unclear whether migraine can cause BMS.”
Introduction
Epidemiological data on burning mouth syndrome and migraine are lacking here. In addition, data on the association between psychiatric disorders and burning mouth syndrome but also migraine should be presented. This is particularly important as it is a relevant factor in both. Furthermore, it should be made clear where the difference between primary and secondary burning mouth syndrome lies. In conclusion, again only a possible association between both disorders should be mentioned.
Answer: Epidemiological information for both diseases was added as follows: “Migraine is one of the most common neurological disorders, characterized by episodes of severe and often unilateral, throbbing, or pulsating headaches associated with multiple symptoms, such as nausea, photophobia, and phonophobia and approximately 15% of the general population has experienced a migraine.” “The overall prevalence of BMS is approximately 4% and it is mainly observed in middle-aged or older adult women and usually presents as an intense burning sensation of the tongue or oral mucosa.”
Answer: We added information regarding the difference between primary and secondary burning mouth syndrome as follows: “Primary BMS is caused by damage to the nerves that control pain and taste, whereas secondary BMS is usually caused by an underlying medical condition.”
Answer: We added further information regarding the possible link between both diseases as follows: “Additionally, common comorbidities of migraine include neck pain and psychological illness such as depression, and anxiety, all of which are leading causes of disability worldwide [2,3]. The pathophysiology of migraine is not completely understood; however, recent studies have proposed that some neuropeptides, such as calcitonin gene-related peptide and substance P, act as mediators [4–6]. Thus, neuropathy may be involved in patients with migraine.” “Although psychological factors and neuropathy are often combined with migraine, to date, no study has investigated the relationship between migraine and BMS.”
Material and methods
Did the classification of BMS and migraine follow ICHD criteria? Please indicate which valid classification was used. Furthermore, there is no description of how the diagnoses were collected from the data and who finally made the diagnosis. There is a lack of differentiation between primary and secondary BMS.
Answer: In this study, the operational definition for migraine and BMS was based on the diagnostic code in the KNHIS. All disease diagnostic codes were identified using the Korean Classification of Disease, fifth edition (KCD-5) modification of the International Classification of Disease and Related Health Problems, 10th revision (ICD-10). South Korea has maintained a nationwide health insurance system since 1963 under the Korean National Health Insurance Service (KNHIS), and nearly all of the data in the health system are centralized in large databases. The KNHIS contains all medical costs among beneficiaries, medical facilities, and the government. Almost all medical data, including diagnostic codes, procedures, prescription drugs, and personal information, are included in the KNHIS database. No patient’s health care records are duplicated or omitted, because all South Korean residents receive a unique identification number at birth. The present study used the database for KNHIS-NSC 2002 to 2013, which comprised 1,025,340 nationally representative random individuals, accounting for approximately 2.2% of the South Korean population in 2002. Stratified random sampling was performed using 1476 strata, by age (18 groups), sex (2 groups), and income level (41 groups: 40 for health insurance beneficiaries and 1 for medical aid) among the South Korean population of 46 million in 2002. However, as you commented, we could not differentiate between primary and secondary BMS because this study was based on diagnostic codes. Thus, we added this as a limitation in the section of the Discussion.
Why were 852 patients with migraine excluded, this is only shown in Figure 1, but not justified in the text.
Answer: In this study, we described the inclusion and exclusion criteria for the operational definition of migraine. Among those, we conducted a washout period of one year (2002) in this study to remove the possibility of BMS diagnosis prior to migraine diagnosis during the index period (2003-2005). During this process, we detected 852 patients diagnosed with migraine in 2002 who were excluded, accordingly. To improve clarity on this point, we modified Figure 1.
In addition, it should be explained why the CCI was used as a score for comorbidities, since it does not include most comorbid diseases that are relevant for burning mouth syndrome and migraine (e.g. rheumatic diseases).
Answer: In this study, we used CCI to adjust for comorbidity. As you are aware, CCI is a weighted index to predict the risk of death within 1 year of hospitalization for patients with specific comorbid conditions and it also includes several rheumatologic diseases (M05.x, M06.x, M31.5, M32.x-M34.x, M35.1, M35.3, M36.0). Additionally, several previous studies showed that CCI is an efficient approach for risk adjustment from administrative databases (J Clin Epidemiol . 1996 Dec;49(12):1429-33 / Methods Inf Med. 1993 Nov;32(5):382-7. J Crit Care. 2005 Mar;20(1):12-9 / J Clin Epidemiol. 2000 Dec;53(12):1258-67).
Results
It would also be useful to present the prevalence as well as incidence of burning mouth syndrome in the entire cohort. It would be useful to present the significant results in particular.
Answer: We agreed with your recommendation. However, in this representative cohort dataset, we could only identify the disease status based on diagnostic codes entered when patients visited the clinics. Hence, we could not assess the prevalence and it is pointless to estimate the overall incidence in the entire cohort. These issues are an inevitable limitation of cohort datasets based on claim data.
Discussion
There need not be a direct association between burning mouth syndrome and migraine; it is also possible that psychiatric disorders (depression and anxiety), which are very common and increase with age, are responsible for the suspected association. The part on the association between migraine and burning mouth syndrome regarding systemic inflammation is not understandable
Answer: In the present study, our hypothesis is that psychological factors and neuropathy may be the possible mechanisms for the association of migraine with subsequent BMS development.
Thus, to improve clarity, we thoroughly modified the Discussion section in terms of this hypothesis.
On the strengths of the study, here it is just unclear how the diagnosis of burning mouth syndrome of migraine was made in each case. Who made the diagnosis should be presented in the methods. Also here it is not clear why the CCI was selected and which connection should Answer: In this claim data-based cohort study, we identified the diagnosis of migraine and BMS based on the ICD-10 diagnostic code, not medical records that include details such as the patients’ medical history, the results of neurocognitive questionnaires, and laboratory results. This methodology inevitably entails the possibility of under- or over-diagnosis, which means that this study has a misclassification bias. To overcome this problem, we enrolled only patients with migraine diagnosed by neurologists and BMS diagnosed by otorhinolaryngologists. Additionally, in this study, we used the Charlson Comorbidity Index (CCI) to estimate health outcomes measured using the ICD-10 code. The CCI accounts for multiple comorbidities by creating a sum score weighted according to the presence of various conditions. Previously, various studies demonstrated that the CCI has been validated for various diseases in longitudinal studies.
Round 2
Reviewer 3 Report
The colleagues answered all questions in detail.